# Antimicrobial Multidrug Resistance: Clinical Implications for Infection Management in Critically Ill Patients

**DOI:** 10.3390/microorganisms11102575

**Published:** 2023-10-16

**Authors:** Gamze Kalın, Emine Alp, Arthur Chouaikhi, Claire Roger

**Affiliations:** 1Department of Infectious Diseases and Clinical Microbiology, Faculty of Medicine, Erciyes University, Kayseri 38280, Türkiye; 2Department of Infectious Diseases and Clinical Microbiology, Faculty of Medicine, Ankara Yıldırım Beyazıt University, Ankara 06760, Türkiye; eminealpmese@gmail.com; 3Department of Anesthesiology and Intensive Care, Pain and Emergency Medicine, Nîmes-Caremeau University Hospital, Place du Professeur Robert Debré, CEDEX 9, 30029 Nîmes, France; arthur.chouaikhi@outlook.fr; 4UR UM 103 IMAGINE, Faculty of Medicine, Montpellier University, Chemin du Carreau de Lanes, 30029 Nîmes, France

**Keywords:** antimicrobial resistance, ICU, sepsis, difficult to treat pathogens

## Abstract

The increasing incidence of antimicrobial resistance (AMR) worldwide represents a serious threat in the management of sepsis. Due to resistance to the most common antimicrobials prescribed, multidrug-resistant (MDR) pathogens have been associated with delays in adequate antimicrobial therapy leading to significant increases in mortality, along with prolonged hospital length of stay (LOS) and increases in healthcare costs. In response to MDR infections and the delay of microbiological results, broad-spectrum antibiotics are frequently used in empirical antimicrobial therapy. This can contribute to the overuse and misuse of antibiotics, further promoting the development of resistance. Multiple measures have been suggested to combat AMR. This review will focus on describing the epidemiology and trends concerning MDR pathogens. Additionally, it will explore the crucial aspects of identifying patients susceptible to MDR infections and optimizing antimicrobial drug dosing, which are both pivotal considerations in the fight against AMR. Expert commentary: The increasing AMR in ICUs worldwide makes the empirical antibiotic therapy challenging in septic patients. An AMR surveillance program together with improvements in MDR identification based on patient risk stratification and molecular rapid diagnostic tools may further help tailoring antimicrobial therapies and avoid unnecessary broad-spectrum antibiotics. Continuous infusions of antibiotics, therapeutic drug monitoring (TDM)-based dosing regimens and combination therapy may contribute to optimizing antimicrobial therapy and limiting the emergence of resistance.

## 1. Introduction

Antimicrobial resistance (AMR) is an important factor affecting clinical responses in intensive care units (ICUs) patients. Although the benefits of antimicrobial therapy are well known, potential harms are often underestimated. The development and spread of antibiotic resistance, which was declared a global health problem and requires an urgent action plan by the World Health Organization (WHO), causes great difficulties in the treatment of nosocomial infections [1]. Nosocomial infections are a significant threat in ICUs. They are one of the important causes of morbidity and mortality for patients. According to the data from the European Center for Disease Prevention and Control (ECDC), at least one episode of nosocomial infection may be developed in 8% of patients followed for longer than 48 h in ICUs [2]. Among the reasons for the high rates of AMR is the increased use of different antibiotics in ICUs. AMR and limited available treatment options pose challenges in the management of nosocomial infections. Especially in ICUs, the duration of hospitalization is prolonged due to antimicrobial failure, and this situation reveals a pattern of increasing costs [3]. The unnecessary antibiotic use can be reduced as a result of strategies developed for the control of resistance in ICUs. Establishing policies for the effective use of antibiotics requires regular epidemiological surveillance and a multidisciplinary approach [1,3]. As part of these strategies, achieving precise predictions regarding which patients are at a higher risk of colonization or infection by MDR pathogens is a pivotal breakthrough in combating AMR in critically ill patients. In addition, the recent development of advanced microbiological tools has significantly expedited the identification process, allowing for informed decision making in the early stages of nosocomial infections. This enables the timely selection of an appropriate empirical antibiotic treatment regimen, thereby avoiding unnecessary antibiotic overuse. Once the antimicrobial therapy is chosen, it becomes imperative to establish optimal drug dosing and administration protocols to ensure both efficacy and the prevention of resistant pathogen emergence. In this review, we will describe the epidemiology and trends of MDR pathogens, then focus on the strategies for the identification of MDR pathogens and finally provide insights for antimicrobial therapy optimization against MDR infections and in the presence of limited resources.

## 2. Epidemiology and Trends in AMR

### 2.1. Epidemiology

Although it was seen as a scientific curiosity rather than a clinical problem in the first years of the antibiotic era, there have been some structural changes in bacterial populations with the selective killing of susceptible microorganisms as a result of the widespread use of antibacterial agents. This contributed to the increase in AMR. Due to the spread of AMR on a global scale, the effectiveness of antimicrobial therapy is gradually decreasing in every corner of the world, which leads to more difficult-to-treat infections and thus to deaths [4]. AMR is a serious public health problem, causing at least 1.27 million deaths worldwide annually and reportedly linked to more than 5 million deaths in 2019. In addition, it was reported that infections caused by 2.8 million resistant microorganisms are seen in the U.S.A. every year [5]. In the Antibiotic Resistance Threat Report published in 2019, it was stated that more than 35,000 deaths were associated with AMR. In addition, in the U.S.A., it was stated that the deaths reached 48,000, with the addition of deaths due to Clostridium difficile diarrhea associated with antibiotic use [4,5]. Globally, it is estimated that AMR-related deaths exceeded 1.2 million in 2019, and if adequate measures are not taken, AMR might cause approximately 10 million deaths per year by 2050 [6]. In the study conducted by Sader et al., when the ICU and other departments of the same hospitals in the U.S.A. and Europe were compared, the prevalence of high AMR was shown in ICUs [7]. Within the scope of the SENTRY Study, a total of 23.233 isolates, 5.989 of which were in ICUs, were collected over a 3-year period, and antimicrobial susceptibility was evaluated by liquid microdilution methods. The most common Gram-negative bacteria (GNB) among these isolates were included. When the sensitivity rates were compared, it was found that the sensitivity rates were lower in ICUs [7]. In a study from Far East Asia, the AMR rates were examined in patients followed up with the diagnosis of ventilator-associated pneumonia (VAP) [8]. The most frequently isolated pathogens were *Acinetobacter* spp., *Klebsiella* spp. and *Pseudomonas aeruginosa*. For *Acinetobacter* spp., colistin resistance was found in 1.5% of the isolates, cefoperazone/sulbactam resistance in 4.3%, and resistance to other agents in 70%. In *Klebsiella* spp., high resistance was found to other antibiotics (cefoperazone, ceftazidime, ceftriaxone, cefepime and levofloxacin) except for amikacin (5.1%) [8]. Carbapenem and quinolone resistance was found in over 80% of *Pseudomonas aeruginosa* isolates and was thought to be associated with the widespread use of quinolones and cephalosporins in ICUs [8]. It is also known that *Klebsiella pneumoniae* is one of the most common causes of nosocomial infections in ICUs, and multi-drug-resistant (MDR) strains have posed a serious risk in recent years. In the study of Sharma et al., the AMR rates of *Klebsiella* spp. isolated from endotracheal aspirates between 2018 and 2022 were compared [9]. It was determined that Extensively Drug-Resistant (XDR) infections increased from 62.5% in 2018 to 71% in 2022 among patients who underwent mechanical ventilation in the ICU. The AMR rates were determined as 90% for amoxicillin/clavulanate, 100% for ciprofloxacin, 92.5% for piperacillin/tazobactam and 95% for cefoperazone/sulbactam in 2018. It was also shown that no strain was susceptible to these antibiotics in 2022 [9]. In a study conducted in Italy, a total of 113,635 bacterial isolates were tested between 2019 and 2022, and AMR was found in 11,901 isolates. It demonstrated an increase in carbapenem-resistant isolates from 2.62% to 4.56%, in vancomycin-resistant Enterococci (VRE) from 0.58% to 2.21% and in methicillin-resistant Staphylococcus aureus (MRSA) from 1.84% to 2.81% [10]. In recent years, AMR was shown to affect the mortality rates of patients in ICUs. Dautzenberg et al. found that patients colonized by carbapenem-resistant *Enterobacterales* had a higher mortality rate (40%) compared to patients infected by carbapenem-susceptible strains (28%) in ICUs [11]. In a study evaluating the prognostic importance of *Klebsiella* spp. in the ICU, the mortality rates of infections caused by carbapenem-resistant *Klebsiella pneumoniae* were found to be 20% higher than those of infections due to carbapenem-susceptible strains. This emphasizes the relationship between the presence of resistant microorganisms, which are defined as colonization or infection agents, and mortality [12]. In some studies, it was noted that in the treatment of carbapenem-resistant *Klebsiella pneumoniae* infections, the 30-day mortality increased if adequate microbiological eradication was not achieved on the 7th day [13]. Among the microorganisms in bacteremia cases reported from 49 countries in the Global AMR and Usage Surveillance System (GLASS) reports received in 2019 within the scope of a sustainability project, the incidence rate of MRSA was 12.11% (IQR 6.4–26.4), while the incidence rate of *Escherichia coli* resistant to third-generation cephalosporins was 36.0% (IQR 15.2–63.0) [14]. The antimicrobial susceptibility results were evaluated in urinary tract infections caused by GNB within the scope of the AMR Trends Monitoring Study (SMART), a multinational surveillance program conducted in 2009. Of the 2135 isolates, 48.9% were *Escherichia coli*, 14.5% *Klebsiella pneumoniae*, 6.4% *Proteus mirabilis*, 4.6% *Enterobacter cloacae*, and 2.5% *Klebsiella oxytoca*. Extended-Spectrum Beta-Lactamase (ESBL) production was seen in only 7% of the cases. The resistance to carbapenem was found to be higher in ESBL (+) *Klebsiella pneumoniae* (68.8% imipenem-susceptible) compared to *Escherichia coli* strains (100% imipenem-susceptible) [15]. However, in a study from Southeast Asia, the rate of ESBL-producing *E. coli* was 43.5%, and that of non-ESBL-producing *E. coli* was found to be 56.4% in patients followed up with urinary tract infections. The antimicrobial resistance rates are higher for ESBL-producing bacteria than for non-ESBL-producing bacteria. No carbapenem resistance strain was found among ESBL-producing strains. The ESBL-producing *K. pneumonia* rate was 23.6%, and that of non-ESBL-producing *K. pneumonia* was 76.8%. Carbapenem resistance was found to be 2.4% [16]. In a study by Chakraborty et al., in which the authors defined the antibiotic resistance patterns of causative pathogens in adult and pediatric patients in the ICUs of a tertiary hospital, it was found that GNB showed high resistance to different antibiotic classes [17]. In the adult age group, the resistance of *Escherichia coli* to second-generation cephalosporins 95%, beta-lactams 95%, aminoglycosides 75%, fluoroquinolones 95% and carbapenems 75% was reported. It was shown that 75% of *Klebsiella* spp. were resistant to aminoglycosides, 100% to second-generation cephalosporins, 94% to beta-lactams, 92% to fluoroquinolones, and 88% to carbapenems. *Proteus* spp. were found to be 100% resistant to cephalosporins. *Acinetobacter baumannii* also showed high resistance to all antibiotic classes. Also, *Pseudomonas* spp. showed 100% resistance to third-generation cephalosporins, 79% resistance to fourth generation cephalosporins, 74% resistance to beta-lactams, 74% resistance to fluoroquinolones and 89% resistance to carbapenems. *Enterococcus* spp. showed 96% resistance to ampicillin and 98% resistance to fluoroquinolones [17]. Especially, the mortality rates may also increase due to AMR in critical hospital units. In a meta-analysis that investigated the relationship between mortality and multidrug-resistant *Pseudomonas aeruginosa* (MDRPA)/SPM-1-producing strains, 3201 cases of *Pseudomonas aeruginosa* infection were included. It was shown that patients infected with MDR-PA (44.6%) had a higher mortality rate than patients without resistant infections (24.8%). In addition, it was stated that the mortality of patients infected with non-SPM-1-producing strains was four times higher than that of patients infected with SPM-1-producing strains, but the difference was not statistically significant [18]. In a retrospective study conducted in an ICU in Japan, the clinical features and mortality rates of critically ill patients followed up for bacteremia caused by ESBL-producing and non-ESBL-producing *Escherichia coli* were evaluated. The success rate of appropriate empirical antibiotic therapy was lower (54.2% vs. 96.1%; *p* < 0.01) for ESBL (+) *Escherichia coli* bacteremia, and the mortality rate was found to be two-fold higher (37.5% vs. 15.6%, respectively; *p* = 0.04). However, when the analysis was limited to patients receiving appropriate empirical treatment, there was no significant difference [19]. Among the causes of AMR, which was among the effective causes of this increase in mortality, inappropriate empirical antibiotic treatments and delays in therapy initiation were held responsible [12,20]. It is only a matter of time before microorganisms carrying antibiotic-resistant genes spread in ICUs and cause infections through factors such as cross-contamination, environmental contamination and microbiota alterations, and this is quite worrying. Therefore, taking the necessary infection control measures plays a key role against the establishment of these microorganisms and in the prevention of the spread of resistance genes. In a meta-analysis evaluating the AMR rates in ICUs in low- and middle-income countries, the mortality rates in low-income countries (33.6%) were much higher than in high-income countries (<20%). MDR Gram-negative strains (*Acinetobacter baumannii*, 24%, *Pseudomonas aeruginosa*, 16% and *Klebsiella pneumoniae*, 15%) played a more dominant role in low-income countries than in high-income countries. While the sum of the AMR rates for these three species was only 25% in Western European countries, Gram-positive pathogens were more prominent. As a result, the selection pressure caused by the excessive consumption of antibiotics and the absence of barriers to the spread of resistant strains in the hospital environment were thought to be among the determinants of this AMR [21].

### 2.2. Trends

The increasing antibiotic resistance in ICUs is becoming a concern day by day. Microorganisms that are difficult to treat cause serious conditions that may result in mortality due to infection after colonization. Therefore, in addition to the development of new treatment options, control policies and emergency action plans should be established for the careful use of antibiotics [22]. Although MRSA and VRE, which are Gram-positive pathogens, were seen as important risk factors in many hospitals and ICUs in the past, they have recently ceased to be a problem compared to other resistant microorganisms. For example, the prevalence of MRSA in intra-abdominal infections was reported to be 1% in an abdominal sepsis study [23]. The prevalence of hospital-associated MRSA infections declined significantly between 2005 and 2017, from 114 to 94 cases per 10,000 hospitalizations. These infections remained stable in 2017 and increased again in 2019 during the COVID-19 pandemic [24]. It was reported that globally, the rate of MRSA infections increased to 44.2% in 2008 and has decreased since then [19]. In the study of Falagas et al., it was shown that the prevalence of MRSA decreased from 36% to 24% in South Africa between 2007 and 2011 [25]. In recent years, new cephalosporin-derived drugs (such as ceftaroline, ceftazidime/avibactam, and ceftolozane/tazobactam) showed improved activity in the fight against Staphylococcus aureus infections. Studies were conducted on the combination of ceftaroline with daptomycin and on ceftaroline monotherapy. Although synergistic combination treatments with other antimicrobial agents were investigated, the presence of resistance to this antimicrobial was described [26]. VRE emerged as an important infectious agent due to rectal colonization, especially in patients receiving long-term antibiotic therapy. In some studies, the prevalence of VRE was reported to be 2.8%, especially in intraabdominal infections [23]. The 2019 AMR Threats Report stated that there were significant decreases (from 84,800 cases to 54,500 cases) in community and nosocomial cases associated with VRE, which is considered a serious threat, in 2017 compared to 2012 [27]. Bacteremia due to VRE is also common in ICUs. Nosocomial infections due to these resistant pathogens lead to higher costs compared to infections caused by antibiotic-susceptible agents. The treatment success rates are also declining due to acquired resistance to ampicillin, penicillin, aminoglycosides and glycopeptides. However, daptomycin and tigecycline are among the treatment options that still maintain their activity.

Other recently identified pathogens that pose a serious threat in ICUs are ESBL-producing *Enterobacterales*, MDR *Pseudomonas aeruginosa*, carbapenemase-producing *Enterobacterales*, carbapenem-resistant *Enterobacterales* (CRE) and *Acinetobacter* spp. Especially, MDR GNB (MDR-GNB) is perceived as an important threat [1]. It was determined that there are great geographical differences between species in terms of incidence of MDR-GNB. In a large multicenter international cohort of BSI, the EUROBACT 2 cohort, carbapenem resistance was present in 37.8% of *Klebsiella pneumoniae* isolates, 84.6% of *Acinetobacter baumannii* isolates, 7.4% of *Escherichia coli* isolates and 33.2% of *Pseudomonas aeruginosa* isolates. Difficult-to-treat resistance (DTR) was present in 23.5% of the isolates, and pan-drug resistance in 1.5% [28]. In the Abdominal Sepsis Study (ABSES), the resistance rate in intraabdominal infections caused by GNB was found to be 9% in Western Europe and 46% in Africa and the Middle East. Carbapenem-resistant GNB were found at a very low rate (0.5%) in Western Europe, while their rate was 15% in Eastern and Southern Europe. It is recommended that these geographical differences should be taken into account in determining the preferred strategies to combat AMR [22]. Although it was stated that MDR-GNB infections are increasing, some studies reported data showing a decrease in problematic pathogens such as MDR *Pseudomonas aeruginosa* or *Acinetobacter* spp. In a study by Furuya et al., it was determined that there was a 6–7% decrease in MDR microorganisms in patients hospitalized in the ICU between 2006 and 2014 [29]. This decrease was associated with hospital-wide infection control and prevention [29]. Italian researchers illustrated that interventions to minimize the transmission of carbapenemase-producing microorganisms such as *Klebsiella pneumoniae*, Acinetobacter and MDR *Pseudomonas aeruginosa* reduced the resistance rates from 91% to 13% [30].

## 3. Predicting MDR Infections in Critically Ill Patients

Infections caused by multidrug resistant (MDR) pathogens are associated with the risks of inadequate empirical antimicrobial therapy on one hand and of unnecessary broad-spectrum coverage on the other hand. However, predicting infections caused by MDR pathogens remains highly challenging. In a large recent U.S. cohort study, the net prevalence of at least one Gram-positive pathogen or at least one GN pathogen was 13.6% and 13.2%, respectively, leading to the frequent administration of unnecessary broad-spectrum, empiric antibiotics [31]. Additionally, different studies attempted to analyze the impact of MDR-GNB infections on clinical and economic outcomes and uniformly reported worse outcomes for MDR-GNB infections compared to episodes due to non-MDR-GNB [12,32,33]. This justifies the interest to identify risk factors for this condition to guide empirical therapy before the availability of culture results and limit the prescription of unnecessary broad-spectrum antimicrobials. To minimize the resistance development, the stratification of patients at risk for MDR-GNB is crucial but unfortunately remains challenging in the daily practice. Thus, clinical tools to assess the risk of MDR infections are needed to optimize empiric antimicrobial therapy. Ensuring optimal outcomes for patients infected with MDR pathogens requires a multifaceted approach, including appropriate risk stratification, knowledge of local resistance, as well as the integration of diagnostic tools and antimicrobial stewardship (AMS) interventions.

### 3.1. Multidrug Resistance Risk Factors

Multiple factors contribute to the onset of MDR pathogens; so, for these reasons, an accurate and early identification of the infectious pathogen is required to implement the appropriate treatment that may impact mortality. Emerging MDR-GNB are related both to individual risk factors and to local epidemiology, including community, long-term care facilities or hospital setting. While ESBL *Enterobacterales* are found in the community and hospital environment, CRE, MDR *Pseudomonas aeruginosa* and MDR *A. baumannii* are still mainly found in hospitalized patients. As described previously, given the great heterogeneity of diseases and variations in MDR GNB distribution worldwide, the risk factors and prevalence of MDR GNB widely vary. Several prospective and retrospective studies aimed to report the risk factors for developing infections caused by MDR pathogens [34]. In a case–control study focusing on the risk factors for MDR-GNB carriage upon admission, liver cirrhosis (OR 6.54, 95% CI 2.17–19.17, *p* < 0.01), previous MDR-GNB carriage (OR 5.34, 95% CI 1.55–16.60, *p* < 0.01), digestive surgery in the last year (OR 2.83, 95% CI 1.29–5.89, *p* < 0.01) and hospital length of stay (LOS) in the last year (OR 1.01 per each additional day, 95% CI 1.00–1.03, *p* = 0.03) were identified as independent risk factors [34]. In a meta-analysis of eighteen studies, male gender (OR 1.40, 95% CI 1.09–1.80), having an operative procedure (OR 1.31, 95%CI 1.10–1.56), a central venous catheter (OR 1.22, 95%CI 1.01–1.48), mechanical ventilation (OR 1.25, 95%CI 1.07–1.46), previous antibiotic therapy (OR 1.66, 95%CI 1.41–1.96), ICU LOS (weighted mean difference 8.18, 95% CI 0.27–16.10) and types of health-associated infections (HAI) were the identified risk factors for MDR-GNB infection in the ICU [35]. For pulmonary infections, three major risk factors were mostly reported: previous antibiotic exposure, with the nature of the antibiotic itself influencing the emergence of specific MDR organisms, hospital and ICU LOS and prior colonization with MDR pathogens [36]. Indeed, the colonization pressure appears as an independent risk factor for acquiring MDR pathogens in the ICU [37]. In a multicenter study, the number of additional colonization sites was a significant independent risk factor (OR, 3.37 per site; 95% CI, 2.56–4.43; *p* < 0.0001) for carbapenem-resistant *K. pneumoniae* BSI development among carbapenem-resistant *K. pneumoniae* rectal carriers [38].

As no single risk factor may reliably predict MDR infections, different scores integrating multiple risk factors have been developed.

### 3.2. Multidrug Resistance Risk Scores

Clinical prediction models may be used for identifying critically ill patients at risk of infection with MDR pathogens. In a single-center study, the authors developed a risk-assessing tool to predict carbapenem-resistant etiology in critically ill patients with BSI admitted to the ICU [39]. This score includes age, presence of sepsis, previous cardiovascular surgery, SAPS II, rectal colonization and invasive respiratory infection by the same pathogen and show good predictive performance. The Giannella score may be useful to predict carbapenem-resistant *Klebsiella pneumoniae* bacteremia. It ranges from 0 to 28 and was developed based on four independent variables (admission to the ICU, abdominal invasive procedure, chemotherapy/radiation therapy and number of additional colonization sites). At a cut-off of ≥2, the model exhibited excellent sensitivity and negative predictive value (>90%) [37]. This score was externally validated in a Spanish cohort with an AUROC of 0.92 (95% confidence interval, 0.87–0.98). The optimal cutoff point was fixed at <7 and ≥7 with 92.9% sensitivity and 84.8% specificity [40]. For MDR *Pseudomonas aeruginosa*, the risk score integrates previous MDR *Pseudomonas aeruginosa* isolation (11 points), prior antibiotic use (3 points), hospital-acquired infection (2 points) and septic shock at diagnosis (2 points) as the main determinants for developing MDR *Pseudomonas aeruginosa* infections [41]. By using a cut-off value of 7 points (theoretical risk of 56%) the sensitivity increased to 94.5% (with a specificity of 41%).

Clinical decision trees have also been proposed to estimate the likelihood of infection with MDR pathogens. For ESBL-producing bacteria in bacteremic patients, a clinical decision tree based on five predictors, i.e., history of ESBL colonization/infection, chronic indwelling vascular hardware, age ≥43 years, recent hospitalization in an ESBL high-burden region and ≥6 days of antibiotic exposure in the prior 6 months, showed excellent negative and positive predictive values to predict ESBL-positive bacteremia in a setting with 15% prevalence of ESBL-positive bacteremia [42]. Indeed, it is important to note that the disease prevalence strongly influences the pre-test probability in the context of employing predictive scores or models. It is imperative to duly acknowledge and account for this influence when deploying such tools in specific settings. A resistance frequency rule may further inform empirical antibiotic therapy. Using a large surveillance program, Klinker et al. demonstrated that stratifying patients using the “15% resistance frequency rule” for carbapenem-resistant P. aeruginosa and ESBL *Enterobacterales* could help optimizing an empirical therapy for respiratory tract infections and anticipating the susceptibility testing for newer agents while awaiting the final microbiology results [43].

### 3.3. Rapid Diagnostic Tools (RDT)

Resistance to antimicrobial agents can arise through various molecular mechanisms, which depend on both the specific microorganism and the antimicrobial agent in question. These mechanisms encompass a wide range of genetic events, including the constitutive or inducible expression of resistance genes, as well as the upregulated expression of these resistance genes [44]. Furthermore, some bacteria possess intrinsic resistance to particular types or entire classes of antimicrobial agents. Currently, numerous screening methods are available for characterizing antimicrobial resistance (AMR) in bacterial isolates, utilizing both phenotypic and genotypic approaches. Despite the diverse challenges associated with genotypic detection of mechanisms leading to reduced susceptibility to various antimicrobial agents, molecular techniques are extensively employed in both research and reference laboratories. Among the methods utilized, traditional approaches such as PCR and hybridization techniques have been in use for decades, while newer methods like whole-genome sequencing (WGS) and matrix-assisted laser desorption ionization–time of flight mass spectrometry (MALDI TOF MS) have recently emerged.

#### 3.3.1. Multiplex PCR

The utilization of PCR for monitoring multiple AMR genes within bacteria has been significantly simplified through the adoption of multiplexing techniques, which are now widely used in clinical practice. In a multiplex PCR assay, the concurrent detection of numerous resistance genes is made possible by incorporating distinct primers into the assay mix. These multiplex PCRs are typically designed to target various genes associated with a common resistance phenotype, such as that linked to the prevalent beta-lactamases found in GNB, known for their role in conferring resistance to cephalosporins or carbapenems. By simultaneously screening for these genes, substantial time and effort can be saved to identify the underlying mechanism(s) responsible for the observed resistance phenotype. The impact of mPCR for optimizing the empirical antibiotic therapy was assessed in different observational studies [45,46,47,48,49]. In a single-center study involving fifty-six patients, the mPCR pneumonia panel results could have prompted a change in therapy in 64% patients, with an anticipated mean reduction in time to optimized therapy of approximately 51 h. In addition, the panel identified three cases where antimicrobials should have been altered because the patients were not receiving empiric therapy with activity against the causative pathogen [47]. In another single-center study, the mPCR pneumonia panel allowed for the escalation of the empirical antibiotic therapy in 22% of adult patients with community-acquired pneumonia [46]. The use of the mPCR pneumonia panel for antimicrobial therapy guidance for VAP allowed for the detection of eight MDR pathogens among 100 samples and spared the use of unnecessary broad-spectrum agents [50]. However, better performance was reported when mPCR was integrated into a stewardship program [51,52]. Nevertheless, some limitations remain while using the mPCR pneumonia panel. First, some pathogens (*Enterococcus* spp., *Serratia* spp., *Morganella* spp., *Hafnia alvei*, *Citrobacter* spp., *Stenotrophomonas maltophilia* for the FilmArray pneumonia panel plus (BioFire, Biomérieux), *Acinetobacter baumannii*, *Enterococcus* spp., Streptococcus agalactiae and pyogenes for the Unyvero HPN (Curetis, Unyvero^TM^)) are not included in the panel. A high prevalence of these microorganisms may impact the performance of the mPCR pneumonia panel in determining the appropriate antimicrobial therapy. Second, the over-identification of pathogens is a general concern for all molecular diagnostics. The use of the mPCR pneumonia panel could lead to an unnecessary escalation of antibiotics when multiple microorganisms are identified in clinical samples from typically monomicrobial infections. In observational studies combining mPCR, high clinical suspicion for pneumonia and antimicrobial stewardship, this risk remains low [46,50].

#### 3.3.2. MALDI-TOF MS

Matrix-assisted laser desorption ionization–time of flight mass spectrometry (MALDI-TOF MS) is a technique used to identify bacterial species by analyzing their peptide spectra from MS. The analysis can be performed directly on biological samples in standardized or complex matrices including blood and urine. The “bacterial signature” can then be compared to commercial databases containing species-specific spectral information [53,54]. The rapid implementation of MALDI-TOF MS in microbiology laboratories relies on the rapid and reliable ability to generate pathogen identification results with a high throughput at a low cost. The usefulness of the MALDI-TOF MS technology for rapid bacterial identification directly in blood culture broths has been largely demonstrated, with a significant 1.2–1.5-day reduction in the time for pathogen identification compared to more conventional methods [55,56,57,58,59,60,61]. In addition, its value in AMR determinants identification has been assessed [62,63,64,65]. In MDR GNB infections, the implementation of MALDI-TOF MS combined with AMS achieved improved outcomes by decreasing the time to the administration of appropriate antibiotics for these difficult-to-treat infections by more than two days compared with the pre-intervention study group [61]. In a French single-center study combining MALDI-TOF MS and a chromogenic test (BETA LACTA test^®^) for patients having a positive blood culture with third-generation cephalosporin-resistant bacteria, 64% of the patients had received an inappropriate third-generation cephalosporin therapy as a first-line treatment [66]. In a pre/post quasi-experimental study focused on cases of *A. baumannii* pneumonia and/or bacteremia, the implementation of MALDI-TOF MS in conjunction with AMS interventions yielded positive outcomes. This approach led to a 41-hour reduction in the median time required to initiate an effective therapy compared to the conventional identification method without any interventions. Furthermore, it was associated with a substantial 19% improvement in clinical cure rates and a 2-day decreased LOS during antibiotic therapy [67]. To further use MALDI-TOF MS for characterizing AMR, a semi-quantitative MALDI-TOF-MS-based method for antibiotic resistance detection, MBT-ASTRA^TM,^ was developed based on the comparison of the growth rates of the bacteria cultivated with and without antibiotics. This approach showed interesting results [68,69,70]. In fact, in a study on 89 patients with BSI, an early report with the MBT-ASTRA^TM^ would have saved 112, 40, and 12 days of treatment with third-generation cephalosporins, piperacillin/tazobactam, and carbapenems, respectively, for 39 patients who had not been de-escalated [71]. Several studies highlighted the positive impact of incorporating pharmacist-driven AMS interventions in molecular rapid diagnostic testing [59,62,68]. Despite having the ability to obtain blood culture results up to 1.5 days faster than traditional identification methods, several studies demonstrated that without a real-time AMS intervention, treatment optimization was significantly delayed [58,72].

#### 3.3.3. Value of Antibiograms and Rapid Antimicrobial Susceptibility Testing (rAST)

Antibiograms have advanced significantly in complexity and utility, transitioning from their traditional forms to more contemporary versions capable of delivering increasingly precise and practical antimicrobial susceptibility information [43].

Antibiograms represent an important tool to provide empirical antibiotics recommendations and are a core element of antimicrobial stewardship programs by increasing the likelihood of appropriate initial antimicrobial coverage. Regional cumulative antibiograms have been shown to be feasible and may inform an empirical antibiotic selection for institutions where local surveillance data are missing. They may also be valuable to assist targeted antimicrobial stewardship interventions [73]. The surveillance of AMR in GNB, especially the resistance to all first-line agents, called difficult-to-treat resistance, as well as escalation or combination antibiograms may provide further insights for antimicrobial strategies to treat nonresponding patients [74,75]. An escalation antibiogram can evaluate the susceptibility profile of organisms that are resistant to the drug the patient has been receiving, helping pharmacists determine which drug to switch to. One limitation to the use of antibiograms relies on the turnaround time needed to obtain the required information. The development of rapid AST may help overcome this limitation.

Generally, rapid AST methods can be categorized as either genotypic or phenotypic approaches. Genotypic AST methods identify resistance by screening for specific genetic resistance markers. These methods offer quicker results compared to traditional AST techniques as they do not depend on microbial growth. They can also be conducted directly on biological samples and blood cultures, thereby reducing the turnaround time. However, there are numerous genes associated with resistance, and the current rapid genotypic tests can only detect a subset of them. Furthermore, these tests do not provide insights into the phenotypic expression of resistance genes. Thus, several rapid phenotypic AST systems have been developed, as reported below.

##### Accelerate PHENO^TM^ System

The rapid AST using the Accelerate Pheno^TM^ System provided reproducible clinical results with overall essential and categorical agreements of >94% [76,77,78,79]. Its implementation into routine practice has significantly reduced the time of microbial identification and AST as well as the time for optimal antimicrobial therapy [80,81,82,83,84]. In a study on 27 patients evaluated for the potential clinical impact of the Accelerate Pheno^TM^ System on antimicrobial optimization, 18 (67%) patients could potentially have had their therapy optimized sooner, with an average reduction of 18.1 h in the time to optimal therapy [83]. For BSI, the use of the Accelerate Pheno^TM^ System was also associated with significant decreases in time to antimicrobial de-escalation but did not affect antimicrobial consumption [80,81]. In an RCT comparing rapid ID and phenotypic AST using the Accelerate Pheno^TM^ System to standard of care, the median (IQR) time to antibiotic escalation was faster in the intervention arm for antimicrobial-resistant BSI (18.4 (5.8–72) vs. 61.7 (30.4–72) h; *p* = 0.01) [83]. The best improvement in time occurred when the rapid AST was associated with AMS, even with support limited to an 8 h coverage model [83,85]. The implementation led to a shorter total duration of the antibiotic therapy and the hospital LOS, without an increase in 30-day readmissions for non-critically ill patients (22.1% vs. 14%; *p* = 0.13) [81,84,86]. The addition of real-time notification did not show further improvement over the Accelerate Pheno^TM^ System and an already active ASP suggesting that the Accelerate Pheno^TM^ System can be integrated into healthcare systems with an active ASP even without the resources to include real-time notification [86].

##### REVEAL^®^ AST System

Rapid AST may be performed directly on blood cultures using the REVEAL^®^ automated system (Specific Diagnostics, San Jose, CA, USA). This new system enables the measurement of MICs (minimum inhibitory concentrations) for a wide range of antibiotics and is based on a unique metabolomic signature technology, which detects volatile organic compounds (VOCs) emitted during bacterial growth using standard commercially available 96-well dried antibiotic plates. The antibiogram results are typically obtained within an average of 4.6 h, depending on the bacterial species. One study assessing the performance of the REVEAL^®^ automated system on GNB blood cultures reported an accurate and rapid susceptibility testing with >95% essential categorical agreement with Sensititre and Vitek 2 [87]. One ongoing pre/post quasi-experimental study will evaluate its clinical impact on empirical antimicrobial therapy (NCT NCT05741424); in addition, one RCT will determine the benefit of the BCID2 associated with the REVEAL^®^ automated system for antimicrobial prescription and de-escalation (NCT05909683).

##### dRAST

Another rapid AST system, the dRAST system (QuantaMatrix Inc., Seoul, Korea), showed fast and reliable susceptibility testing directly on monobacterial blood cultures with a major turnaround time reduction (median turnaround time: 6.7 h (range: 4.7–7.9)) [88,89,90,91]. The technique is based on a microfluidic agarose channel system that immobilizes the bacteria in agarose-containing microfluidic chambers. The bacterial growth under different antibiotic culture conditions is tracked by time-lapse imaging [92]. The dRAST system successfully informed infectious disease physicians in the selection of optimal targeted antibiotics for patients with positive blood cultures including polymicrobial infection in >90% of 104 cases receiving non-optimal targeted antibiotics [93].

### 3.4. Future Approaches

Several promising approaches such as whole-genome sequencing, microfluidics, biosensor technologies, isothermal amplification-based NAAT and immunodetection are likely to gain increasing interest in the next future for the rapid diagnosis of AMR [94].

## 4. Optimization of Antimicrobial Therapy in Critically Ill Patients with MDR Infections

### 4.1. Pharmacokinetics Considerations in Critically Ill Patients

During critical illness, in the presence of pathophysiological changes, a drug’s PK may be significantly altered. Firstly, the volume of distribution in these patients is often increased due to several factors, including an inflammatory state induced by sepsis, the presence of a third space (e.g., ascites, pleural effusions) and alterations in the free drug fraction (hypoalbuminemia). Moreover, the antibiotic elimination may be decreased by hepatic or renal organ failures, while it can be increased by augmented renal clearance in patients with traumatic brain injury or multiple trauma [95,96]. These pharmacokinetic changes can lead to underdosing or overdosing, which are associated with clinical and microbiological failures, drug toxicity and resistance emergence [97,98].

### 4.2. Pharmacodynamic Objectives

The pharmacodynamic objectives are mainly based on the assessment of the minimum inhibitory concentration (MIC). The MIC is obtained using an assay that determines in vitro the drug concentration that prevents visible bacterial growth in or on a standardized medium at 34–37 °C using a standardized inoculum. The concentrations used are two-fold variations above and below the concentration of 1 mg/L [99]. The MIC is a determinant of the PK/PD targets used in clinical practice to optimize antimicrobial drug dosing. These targets can be expressed as the ratio of the maximal concentration to the MIC (Cmax/MIC), the free concentration time above the MIC (%fT > MIC) or the ratio of the area under the curve to the MIC (AUC/MIC), depending on whether the antibiotic exhibits dose-dependent, time-dependent or AUC-dependent killing. In critically ill patients, different PK/PD targets for efficacy have been discussed. 100% of time where the free concentration is above the MIC (100% fT > MIC), while some authors recommend a target of 100%fT > 2–5 × MIC [100,101]. The rationale to increase the MIC target is based on tissular diffusion and technical issues about the uncertainty of the MIC and BL measures, as well as on observational studies showing a better microbiological and/or clinical cure [102,103]. By considering MIC-derived objectives, different dosing regimens may be needed. Consequently, in the setting of AMR, higher antimicrobial exposure driven by high MICs may be necessary to achieve the PK/PD targets. Preventing the emergence of resistance is another crucial issue while optimizing drug dosing. Tam et al. demonstrated that during the treatment of *Pseudomonas aeruginosa*, a Cmin/MIC ratio of 1.7 was associated with the emergence of resistant clones, whereas a Cmin/MIC ratio of 6 prevented bacterial regrowth [104]. In the same way, several clinical trials showed that targeting a Cmin/MIC ratio of >4–5 for BL improved the clinical outcomes [105,106,107]. These studies included infections at high risk of clinical or microbiological failure, such as VAP caused by *Pseudomonas aeruginosa* and *Klebsiella pneumoniae*. For MRSA infections treated with vancomycin, an AUC/MIC > 400 mg.h/L has been suggested to avoid treatment failure in strains with an MIC ≤ 2 [108,109].

Finally, the PK/PD studies primarily determine the MIC for bacteria in the planktonic state. However, in infectious diseases, the bacteria can organize into biofilms, which are estimated to be involved in more than 65% of infectious diseases in humans [110]. These biofilms are structured entities constituted of extracellular polymeric substances such as polysaccharides, proteins and extracellular DNA [111]. They confer resistance to the bacteria through various mechanisms, including slowing down the cell cycle, horizontal transfer, secretion of enzymes that hydrolyze or bind antibiotics and decreased antibiotic diffusion [112]. Many techniques are available for studying biofilms in research (optical or electron microscopy, crystal violet staining or the Calgary system). Some techniques have emerged in recent years allowing for the qualitative or quantitative analysis of biofilms (e.g., BioFlux 200 system—Fluxion Biosciences Inc., Alameda, CA, USA, Antibiofilmogram, Biofilm Control, St Beauzire, France). Numerous research applications are possible, including describing bacterial cooperation within a biofilm, observing biofilm development on different interfaces or media and measuring the effectiveness of an antibiotic therapy in treating established biofilms It has been suggested that persister cells might be one reason for antibiotic treatment [113,114,115,116]. failure and might contribute to the evolution of antibiotic resistance [117]. Biofilm assessment may represent one of the tools to identify and prevent AMR.

### 4.3. Optimization of th Dosing Regimens in MDR Infections

Considering the high risk of underdosage and the highest PD target required, the optimization of antimicrobial drug dosing in critically ill patients appears essential.

#### 4.3.1. Prolonged/Continuous Infusion of Beta-Lactams

The prolonged IV administration of antibiotics can increase the time above the MIC target [118,119]. For example, the prolonged infusion of meropenem more frequently achieved the 90%fT > MIC target, especially in the treatment of *Klebsiella pneumoniae* infections when the MIC was greater than 2 mg/L (breakpoint: 8 mg/L) compared to a 30-min infusion (98–99% vs. 61–83%) [120]. Furthermore, the continuous administration of BL such as piperacillin/tazobactam and meropenem showed a reduction in clinical failures compared to an intermittent administration [121,122]. These studies mainly focused on sepsis and VAP, regardless of the documented bacteria. These findings highlight the importance of a prolonged administration in patients with challenging PK, a high risk of clinical failure or infections with pathogens with high MICs [123].

#### 4.3.2. Therapeutic Drug Monitoring (TDM)

Because of the high PK variability in critically ill patients, an individualized approach to antimicrobial dosing is warranted in this population, especially when difficult-to-treat pathogens are suspected [124]. Various models, such as the use of Monte Carlo simulations, have been developed to better understand the PK of antimicrobial drugs in a given population and to predict the likelihood of achieving pharmacokinetic/pharmacodynamic (PK/PD) targets based on the administered and simulated dosing regimen [125]. Recently, complex mathematical modelling and PK models have been embedded into dosing software to assist with drug dosing [126,127]. In addition, repeated therapeutic drug monitoring (TDM) can confirm the achievement of the PK/PD targets for an individual patient [128,129]. Dosing software implementing Bayesian method combines a given population PK model with the data from an individual patient to determine the optimal dose adjustment. Such approach has shown an increased frequency of target attainment for several antimicrobial agents and aims to better understand each patient’s unique pharmacological profile [127,130,131,132]. However, the first RCTs comparing TDM-guided dosing adjustment combined or not to dosing software failed to demonstrate a significant clinical impact [130,133]. The PK/PD targets for efficacy have been the first concern for TDM use. However, TDM is also crucial for preventing unnecessary and potentially harmful high antibiotics concentrations, which could result in toxic effects. Recent research has highlighted the importance of TDM in reducing the risk of non-BL antimicrobial-related toxicity [134]. In addition, in a retrospective study involving 93 patients, there was no evidence of excessive drug toxicity associated with doses exceeding the recommended levels based on TDM, whether for meropenem or piperacillin/tazobactam, even though the daily doses in the high-dose groups exceeded the standard regimen by more than 40% [135]. Nevertheless, the primary obstacle to adjusting dosages based on TDM to minimize toxicity lies in the absence of well-established BL concentration thresholds, making it imperative to invest efforts to establish toxicodynamic targets. Finally, PK/PD targets for preventing the emergence of resistance have been considered to optimize the antimicrobial therapy [136]. The mutant prevention concentration represents the antibiotic level needed to inhibit the growth of initial mutant bacteria that can selectively thrive at concentrations exceeding the MIC. The antibiotic concentration range between the MIC and the mutant prevention concentration is referred to as the mutant selection window. Antibiotic concentrations within this mutant selection window encourage the growth of antibiotic-resistant bacterial pathogens. Consequently, to effectively prevent the emergence of resistance, antibiotic exposure should be sustained at levels above the mutant selection window. However, there remains a lack of research to define the target antibiotic exposures needed to minimize the development of resistance.

#### 4.3.3. Combination Therapy

The administration of combination therapy for MDR GNB is a controversial issue, with studies leading to conflicting conclusions [137]. Faster bacterial clearance, prevention of the development of bacterial resistance and synergistic or additive effects have been advocated to support the combination therapy, while potential side effects like increased toxicity and higher costs are possible drawbacks. Although increased bactericidal activity and lower re-growth rates were reported in vitro for colistin/fosfomycin and polymyxin/rifampicin against *K. pneumoniae* and for imipenem/amikacin or imipenem/tobramycin against *Pseudomonas aeruginosa*, meta-analyses on in vivo combination therapy for MDR GNB demonstrated positive outcomes only for patients with bloodstream infections due to carbapenemase-producing *Enterobacterales* [138,139]. Other potential benefits of a combination therapy for treating MDR pathogens rely on carbapenem sparing and the prevention of resistance by persistent cell killing. A recent study demonstrated, in a hollow fiber model, that both the piperacillin/tazobactam/amikacin combination and a meropenem monotherapy regimen attained a rapid ESBL *E. coli* killing within 24 h and did not result in the emergence of resistant subpopulations [140]. In the presence of *A. baumannii*, the combination of colistin and amikacin was most effective for the eradication of persister cells [141].

## 5. Conclusions

To address the impact of AMR, the healthcare systems need to prioritize measures such as national and local AMR surveillance programs as well as infection prevention control. Additionally, an early and accurate diagnosis of the causative pathogen and the establishment of its resistance profile are crucial for tailoring the antibiotic therapy to individual patients, optimizing the treatment outcomes and minimizing the spread of drug-resistant infections. New rapid diagnostic tools have hastened the time to the identification of MDR pathogens and may help combating the unnecessary broad-spectrum antibiotic overuse.

The optimization of new and old antibiotics administration based on PK/PD understanding and therapeutic drug monitoring is another crucial action to limit the emergence of resistance.

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
