# Peer review of "Antimicrobial Multidrug Resistance: Clinical Implications for Infection Management in Critically Ill Patients"

_microorganisms, 2023, doi:10.3390/microorganisms11102575_

Round 1
Reviewer 1 Report
Good work.
The manuscript needs comments on the following:
The value of antibiograms:
Teitelbaum D. Clin Infect Dis 2022; 75:1763.
Guarascio KJ. J Pharm Pract 2019; 32:19.
Kadri SS. Clin Infect Dis 2018; 67:1803.
None.
Author Response
We would like to thank reviewer 1 for his/her positive comments and valuable suggestions. We have now added the suggested references in the revised manuscript.
Reviewer 2 Report
Review opinion for microorganism 2639089
Gamze Kalın reviewed the antimicrobial resistance impact for critically ill patients. The issue of increasing antimicrobial resistance is a global health problem. Knowing the knowledge of AMR is essential in managing nosocomial infections. I hold positive opinion for the author’s content. However, some of the details should be clarified to make reading more comprehensive.
- Epidemiology an d trends in AMR section, line 89: For Acinetobacter spp., colistin resistance was 1.5%, cefoperazone-sulbactam resistance was 4.3%, and resistance to other agents was 70%. Please mark the reference for the data.
- Section trends, line 181: For example, the prevalence of MRSA in intra-abdominal infections was reported as 1% 181 in the abdominal sepsis study. Also, the reference should be cited.
- Line 189: what is cefazolin-tazobactam? Is this a typing error for ceftolozane-taezobactam?
- Conclusions: This section is not mandatory but can be added to the manuscript if the discussion is To address the impact of AMR, healthcare systems need to prioritize measures such as national and local AMR surveillance programs, infection prevention control. I don’t understand these sentence. The punctuation marks are not used correctly.
- Writing skills and grammar need improvements.
- Writing skills and grammar need improvements.
- Some punctuation marks are not used correctly.
Author Response
We would like to thank reviewer 2 for his/her valuable suggestions.
- Epidemiology and trends in AMR section, line 89: For Acinetobacter spp., colistin resistance was 1.5%, cefoperazone-sulbactam resistance was 4.3%, and resistance to other agents was 70%.Please mark the reference for the data.
Response : the reference has now been added to the revised manuscript.
- Section trends, line 181: For example, the prevalence of MRSA in intra-abdominal infections was reported as 1% 181 in the abdominal sepsis study. Also, the reference should be cited.
Response : Thank you for this comment. The reference has now been added.
- Line 189: what is cefazolin-tazobactam? Is this a typing error for ceftolozane-taezobactam?
Response : We apologize for this typing error. The correction has now been made.
- Conclusions: This section is not mandatory but can be added to the manuscript if the discussion is To address the impact of AMR, healthcare systems need to prioritize measures such as national and local AMR surveillance programs, infection prevention control. I don’t understand these sentence. The punctuation marks are not used correctly.
Response : Thank you for this valuable comment. We have deleted the first sentence and adjusted the punctuation marks in the revised manuscript.
- Writing skills and grammar need improvements.
Response : We do thank the reviewer 2 for this comment. The manuscript has already been checked by our English medical writer. We would appreciate any clarification regarding grammar improvement.
Reviewer 3 Report
I have read your work with keen interest, as it directly pertains to my field of study. While you have conducted a thorough and well-structured review, there are some aspects that I believe could be improved upon.
I am drawn to reference number 15, where remarkably high resistance rates are discussed, which I believe may not be representative of the current reality, at least in our region (data published in India). I kindly request that you review the literature on this aspect.
Regarding the utility of the Film-Array, it might be interesting to discuss its limitations. For example, the omission of certain microorganisms that can cause severe infections in critically ill patients (such as Xanthomonas and Enterococci) or the challenging interpretation of multiple positive results in typically monomicrobial samples (e.g., VAP), could be valuable points of consideration.
Finally, it would be interesting to discuss dosages that allow reaching mutant prevention concentrations and the associated risk of toxicity.
As an anecdotal comment, there is a redundant phrase in line 561.
Author Response
We would like to thank reviewer 3 for his/her positive comments and valuable suggestions to improve the manuscript.
I am drawn to reference number 15, where remarkably high resistance rates are discussed, which I believe may not be representative of the current reality, at least in our region (data published in India). I kindly request that you review the literature on this aspect.
Response: Thank you for this remark. In addition to the SMART study, which includes antimicrobial susceptibility results in microorganisms that cause urinary system infections in the epidemiology section, it was emphasized that the rates vary by adding different studies.
Regarding the utility of the Film-Array, it might be interesting to discuss its limitations. For example, the omission of certain microorganisms that can cause severe infections in critically ill patients (such as Xanthomonas and Enterococci) or the challenging interpretation of multiple positive results in typically monomicrobial samples (e.g., VAP), could be valuable points of consideration.
Response: Thank you for this valuable suggestions. We have commented the points raised in the revised manuscript and added references.
Finally, it would be interesting to discuss dosages that allow reaching mutant prevention concentrations and the associated risk of toxicity.
Response: We fully agree that TDM is valuable for preventing antimicrobial toxicity and emergence of resistance. We have now added a paragraph commenting this point.
As an anecdotal comment, there is a redundant phrase in line 561.
Response: Thank you for this comment. We have now deleted the sentence.
Round 2
Reviewer 3 Report
Dear investigators,
I believe the work has improved sufficiently to be considered for publication in its current form. Congratulations